# Front-of-pack nutritional labels: Understanding by low- and middle-income Mexican consumers

Jorge Vargas-Meza , Alejandra Jáuregui*, Selene Pacheco-Miranda, Alejandra Contreras-Manzano, Simón Barquera

Center Health and Nutrition Research, National Institute of Public Health, Cuernavaca, México

* alejandra.jauregui@insp.mx

## Abstract

Front-of-pack labeling is a cost-effective strategy to decrease population consumption of sodium, sugar, saturated fat, total fat, and trans-fatty acids, considered critical nutrients for chronic disease. Our main objective was to explore the subjective understanding of labels that are currently used internationally among low- and middle-income Mexican consumers. We performed two phases of 10 focus groups with adolescents (13–15 y), young adults (21–23 y), mothers of children 3–12 y, fathers of children 3–12 y and older adults (55–70 y). Seven FOPL were evaluated: Guideline Daily Amounts, Multiple Traffic Light, Chilean Warning labels, Warning labels in Red, 5-Color Nutrition Label, Health Star Rating, and Healthy Choice label. Data was analyzed with a triangulation of researchers using a content analysis, based on three codes: 1) awareness and use of the Guideline Daily Amounts, 2) acceptability, and 3) subjective understanding of labels. Most participants were aware of the Guideline Daily Amounts, however they rarely used it because interpreting the information displayed on the label was too complicated. Health Star Rating, Warning labels, Multiple Traffic Light and the Healthy Choice logo labels were the most understandable, however the acceptability of the The Healthy Choice logo decreased as it did not provide information on specific ingredients. The Warning labels was the only label able to warn about critical nutrients that could represent a health risk. The Warning labels in red was more accepted compared to Warning labels in black. Results show that directive and semi-directive labels, such as Warning labels, Health Star Rating or Multiple Traffic Light, may be better at helping population of low- and middle income make healthier food choices, than non-directive FOPL such as the Guideline Daily Amounts implemented in México. The study results highlight the potential of Warning labels to support decreases in the consumption of critical ingredients in low- and middle-income Mexican consumers.

## Introduction

The inclusion of simplified nutrition information on the front of food packages, known as front-of-pack food labeling (FOPL), is a cost-effective strategy to help consumers make

**Data Availability Statement:** All focus groups transcriptions and quantitative database are available at https://figshare.com/s/7601afd0d47a1c52ab14.

**Funding:** This work was possible thanks to Bloomberg Philantrophies without restrictions.

**Competing interests:** The authors have declared that no competing interests exist.

healthier choices and reduce the risk for chronic diseases at the population level [1–3]. In low- and middle-income countries, FOPL can generate substantial health gains while entirely paying for itself through future reductions of health-care expenditures [4]. This strategy assumes that purchasing behavior can be improved if consumers are presented with a label that is perceived, accepted, understood, and used during food purchase [1].

Obesity and chronic diseases have rapidly become a public health problem in Latin-America [5]. In Mexico, a national epidemiological alert was declared in 2016 due to the high mortality and burden of disease associated with obesity and diabetes [6]. In order to address this public health problem, international agencies such as the World Health Organization, the Pan-American Health Organization and the Food and Agriculture Organization, have recommended decreasing population consumption of sodium, sugar, saturated fat, total fat, and trans-fatty acids [1–3, 7, 8], considered as critical nutrients for chronic diseases [9], and have recognized that an adequate FOPL should contribute to this purpose and the efforts to improve nutrition.

A growing number of countries have implemented FOPL systems, generally based on local evidence on the understanding, acceptability and effect on purchasing intentions of labels [8, 10–12]. In June 2016, Chile introduced a compulsory front-of-pack black warning label (Warning-B) for food products exceeding specified limits of sodium, sugar, energy and saturated fats [13]. In the same year, the Australian government, in collaboration with industry, public health and consumer groups, developed and enforced the Health Star Rating system, rating the overall nutritional profile of packaged foods from 1/2 to 5 stars. In 2017, Nutri-Score labelling, a color-coded scheme associated with letters from A (best nutritional quality) to E (poorer nutritional quality), came into force as a voluntary labelling in France [7]. In Mexico, Guideline Dietary Amounts is the FOPL used by the government to promote healthy dietary choices; however, the selection of this labeling was not based on evidence [10]. Thus, the regulation of FOPL in the country is currently being revised, to replace this system with one that better serves the purpose of decreasing sugar, salt and sodium consumption among the Mexican population.

The effectiveness of a particular FOPL may vary across populations [1]. Studies conducted mostly in European countries suggest that Multiple Traffic Light and Nutriscore may be more effective in increasing the selection of healthier products [7, 14, 15]. In Latin America, studies indicate that Warning labels or logotypes may be more effective than Guideline Daily Amounts or Multiple Traffic Light in helping consumers identify and discourage the consumption of products containing excessive amounts of critical ingredients (i.e., sugar, salt, fat) [16–18]. Although the ultimate goal of FOPL systems is to help consumers identify foods of high nutritional quality, the design of the labels may respond to different purposes. For example, the Health Star Rating and the Nutriscore qualify the overall nutritional quality of a product, but they do not provide specific information on the content of critical nutrients, such as fat, sugar or sodium.

To date, most of the abovementioned studies have focused on quantitative evaluations of the objective understanding of labels or their effect in purchasing intentions [7, 16, 19], without paying much attention to the underlying causes of such effects. Additionally, studies have included highly educated participants,[20, 21] limiting the representativeness of results among lower income and education groups, which are the most representative of the Latin-American populations and generally the most nutritionally at-risk.[22] Although experimental studies provide the best available evidence to evaluate the effectiveness of FOPL, qualitative studies may provide a deeper understanding of the optimal FOPL design among Latin American consumers [23]. Our main objective was to explore the subjective understanding of recent FOPL systems, including the current FOPL in Mexico among low-and middle-income Mexican urban consumers using qualitative methodologies. A secondary objective was to explore the acceptability of the labels.

## Materials and methods

### Study design

We carried out a two-phase study (1st phase in August 2017, 2nd phase in October 2017), with low- and middle-socioeconomic status (SES) residents of Mexico City. Different participants were recruited in each study phase. In both study-phases we aimed to have 10 focus group sessions (5 per SES). The ethical and research committees of the Mexican National Institute of Public Health approved the study protocol and instruments.

An adaptation of the hierarchy of effects model proposed by Grunert and Wills for studying effects of nutrition labels on consumers was the conceptual framework that helped shape the study and instrument design for data collection [1]. This model states that in order for nutrition labels to have any effect, consumers must be exposed to them and must perceive them. The effect will then be mediated by label understanding and acceptability. The acceptability of a label is determined by several factors including whether the label is liked, how attractive it is and the perceived cognitive load of the label [24].

### Participants

To capture the variability of label understanding between population groups we included five groups of participants: adolescents (ages 13–15), young adults (ages 21–23), mothers with children ages 3–12, fathers with children ages 3–12, and older adults (ages 55–70) (S1 Table). We recruited participants at four convenience stores located in low (n = 2) and medium (n = 2) SES neighborhoods, per the Mexican National Institute of Statistics and Geography. Stores were selected by convenience. At each store, we approached shoppers and, after explaining study objectives, we obtained informed consent. Participants were screened for eligibility using a 10-item screener (S1 File) and only those classified as low or medium SES and meeting the characteristics of interest (i.e. groups of participants) were invited to participate in the study and scheduled for a focus group. Exclusion criteria consisted on working (or having a direct relative working) for the food industry or with an education in any health-related area (e.g., nutrition, medicine). A total of 12 participants per group were invited to the focus groups.

### FOPL evaluated

In Phase I, we evaluated seven different FOPLs used internationally (Fig 1): the Mexican Guideline Daily Amounts, Ecuador's Multiple Traffic Light, Chile's Warning labels, the French 5-Color Nutrition Label (a preliminary version of the Nutriscore) [25], the simple version of the Australian Health Star Rating (without the Guideline Daily Amounts), and the international Healthy Choice label. Although some of the original labels contain text indicating approval from a government body; we removed this text to create equivalence across labels [7, 12, 13, 26–28]. In Phase II, we conducted a deeper evaluation of 4 FOPLs: The Health Star Rating, the Multiple Traffic Light, the Warning labels and a red version of the Warning labels (Fig 1). These labels were selected because of their potential to help consumers identify products with high contents of critical nutrients (i.e., Warning labels & Multiple Traffic Light) or because of their high subjective understanding in Phase I (i.e. Health Star Rating).

### Procedures

Focus group sessions took place in facilities with Gesell chambers in central Mexico City. These chambers allowed the research team to observe the focus group through a one-way mirror from a different room without interfering with the focus group dynamic. The same

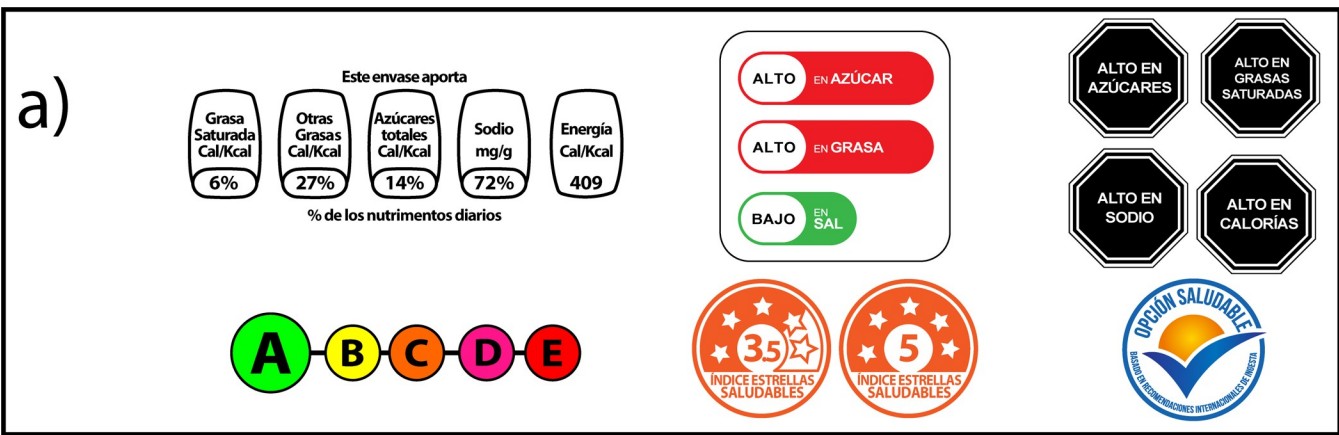

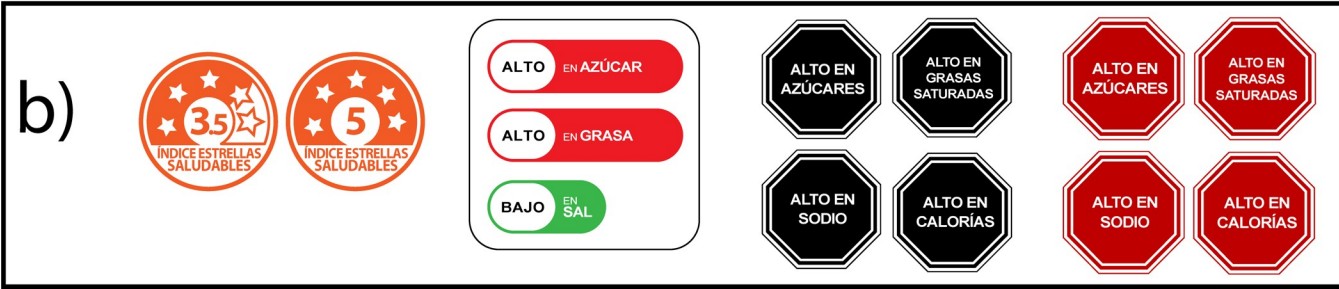

**Fig 1. FOPL utilized in phase I and phase II.** a) Labels used in phase 1: (From left to right) Line 1: Guideline Daily Amount, Multiple Traffic Light, Warning Label from Chile; Line 2: 5-Color Nutrition Label, Health Star Rating and Healthy Choices. b) Labels used in phase 2: (From left to right) Health Star Rating, Multiple Traffic Light, Warning Label from Chile and a red version of the Warning Label.

moderator (male sociologist with more than 30 years of experience conducting qualitative research in food marketing) facilitated all focus groups. Before starting the focus group dynamic, written consent was obtained from all adult participants after explaining the objectives of the session; for minors, written consent from parents and written assent from the minor were also collected. A total of 240 participants were invited to participate in the focus groups, of which 235 presented on the scheduled date (response rate of 97.92%). The first 10 participants to arrive to the focus group entered the Gesell chamber, for a total of 200 participants (100 in each phase). Participants arriving later waited in the lobby for 15 minutes in case any replacement was needed. During the initial stage of the focus groups the facilitator identified two participants who did not meet eligibility criteria (i.e., one participant with medical studies [older adults, low SES] and a nutrition student [young adults, medium SES]). They were replaced by participants waiting in the lobby. After the 15-minute wait, the remaining participants were thanked for their time and received the coupon gift card. Each focus group was audio recorded. The sessions took an average of 2 hours. At the end of the session, participants received a coupon gift card equivalent to $20.8 USD.

**Testing material.** For each FOPL, we placed mock labels on 3 actual food products. These products had sufficiently different nutritional qualities, thus making it possible to rank them by using the labels (except for the Healthy Choices label which was not designed to allow graded ranking) (S2 Table). A different category of food was used for each FOPL, including dairy products, cereals, ready-made products, beverages, and salty snacks. To provide a better view of the FOPL design, labels were also printed on a double-sized board (42.5 x 28 cm) presenting several

grading options of the same label (e.g., 3.5 out of 5 stars for Health Star Rating, 4 different variations of the Warning labels in Black or Red). To simulate the supermarket environment and allow participants to compare each FOPL between products of different categories and nutritional quality, we created a poster of a supermarket shelf for each FOPL (94 x 89 cm).

Before data collection in Phase I, the facilitator tested all instruments (i.e., questionnaire and focus group guide), testing materials and procedures during a pilot focus group with consumers with similar characteristics as our sample. Necessary modifications were done (i.e., it was decided to randomize the order in which FOPL´s were presented to avoid any ordering bias)

**Data collection.** We developed a semi-structured focus group guide addressing the following topics (S2 File): 1) Factors considered when purchasing packaged foods and beverages, 2) Awareness, use, acceptability and subjective understanding of the Mexican Guideline Daily Amounts, and 3) Subjective understanding and acceptability of the remaining six FOPL´s. The order of discussion of the FOPLs (except for Guideline Daily Amounts, which always occurred first) was randomized.

The facilitator started the focus group dynamic with informal interactions with the participants (e.g. asking their names, hobbies, what they do for living, etc.) to establish *rapport*. Then, to get into the topic of the focus group, the facilitator began with the triggering question: *"How do you make decisions when buying packaged foods and drinks sold in supermarkets or convenience stores?"*.

Based on participants' responses, the facilitator ranked the decision-making factors according to the relevance for the group. Because the Guideline Daily Amounts was implemented first as a voluntary label in 2012, and then as the mandatory FOPL in 2015, we were interested in knowing if participants mentioned the label as a decision-making factor when purchasing food products. However, if the group did not mention the Guideline Daily Amounts, the facilitator mentioned the label and investigated the reasons for not using it during the purchase of food. Afterwards, the facilitator explored the use, acceptability and understandability of the Guideline Daily Amounts in greater depth. For this purpose, the label was presented initially using a double-sized board and first impressions and reactions were explored. Then, label understanding and acceptability were further explored by presenting first, a set of actual products and then, the posters with products labeled with the corresponding FOPL (S1 Fig). The facilitator explored the subjective understanding of each FOPL without explaining to participants the intended message of the labels. Label acceptability was explored through label liking, visual attractiveness and perceived cognitive workload [24]. Then, the facilitator moved on to explain the dynamic of the rest of the session exploring the understanding and acceptability of the remaining FOPLs. The same procedure was used as the one for the Guideline Daily Amounts.

## Covariates

We used a self-administered questionnaire to collect demographic information (i.e. age, sex, marital status, and occupation), nutrition knowledge (i.e. awareness of recommended average daily calorie consumption), and use of the Guideline Daily Amounts (i.e. if the person read the label, time to read the label, etc.) (S3 File).

## Data analysis

Descriptive statistics of sociodemographic information were estimated using percentages or means and standard deviations. Differences between low- and middle-SES participants were tested using Chi square tests or t-tests. Quantitative analyses were conducted using Stata version 13.

Audios from focus groups were transcribed clean verbatim. The systematization and interpretation of the information was guided using the content analysis technique using Atlas.Ti

version 11 [29, 30]. We developed a priori central themes for our research questions before data was collected. Additionally, we also developed a code catalog based on transcriptions (S4 File). The analysis was based on three codes: 1) awareness and use of the Guideline Daily Amounts, 2) acceptability, and 3) subjective understanding of labels. Double entry conceptual matrices were developed for each code and participant group (S5 File). Information was encoded and analyzed independently by the focus group moderator, as well as by two additional researchers (JVM and SPM), one of whom attended focus groups as listener. Researchers reached conclusions using the peer review technique, which consists on multiple meetings between researchers to discuss results and their interpretation, until consensus is reached [31–33]. Interpretations reached by researchers are presented in the results section in a narrative form. Since testimonials related with the acceptability and subjective understanding of all FOPL tested were consistent across SES and participant groups, example quotes from the qualitative data, verbatim from the narrative segments, are presented, and identified by SES and participant group, to best represent the concept. Participants' quotes are reported directly as they were spoken, without editing the grammar to avoid losing meaning.

## Results

Participants had a mean age of 34.3 (±16.7) years, were evenly distributed between males and females, almost half were married, and 41.5% had been diagnosed with a chronic disease (Table 1). Almost 80% reported being aware of the Guideline Daily Amounts label and only 19% reported reading the Guideline Daily Amounts when purchasing foods (Table 1). Among those reporting to read the Guideline Daily Amounts during food purchase, 37% considered it as nothing or a little understandable, almost half never or almost never took it into account during food purchasing decisions, and 40% spent more than 30 seconds reading it. No differences were observed between low- and middle-SES participants in most characteristics, however participants with medium SES were more highly educated than participants of low SES, whereas a higher proportion of participants from the low SES had a previous diagnosis of diabetes (p<0.05).

### Mexican Guideline Daily Amounts

Food purchasing decisions were generally based on price, presentation, size, taste and brand of the product, whereas Guideline Daily Amounts was generally not mentioned as a purchasing criterion. This FOPL was only relevant for those individuals that, for health reasons—either their own or of some of their relatives–, were interested in the content of some ingredients that could be of risk. Older adults also tended to report using Guideline Daily Amounts more frequently for health reasons.

Moderator (M): *". . . for example, (Guideline Daily Amounts) labels. . ., ¿do you take them into account . . .?"*

Participant (P): *". . .if I'm on a diet or regime, I do check them . . ."* (Males and females ages 55–70, medium SES)

P2: *"Because of my diabetes, I also check for the sugar."* (Males and females ages 55–70, low SES)

Regarding Guideline Daily Amounts acceptability, participants mentioned that this label was not attractive because the format (i.e. boxes and letters) was too small and the colors (i.e. blue and white) were not pleasant to the eye. As for the subjective understanding, they mentioned that interpreting the percentages and the amounts shown was too complicated; they

**Table 1. Characteristics of participants by socioeconomic status.**

| | Low SES (n = 100) | Medium SES (n = 100) | All (n = 200) |
|---|---|---|---|
| | % | % | n (%) |
| **AGE** | | | |
| 13–17 y | 20.0 | 20.0 | 40 (20.0) |
| 18–54 y | 61.0 | 60.0 | 121 (60.5) |
| 55–70 y | 19.0 | 20.0 | 39 (19.5) |
| **GENDER (Women)** | 50.0 | 54.0 | 104 (52.0) |
| **EDUCATION LEVEL** | | | |
| Primary school or less | 17.0 | 14.0 | 31 (15.5) |
| High School or technical school | 74.0 | 40.0 | 114 (57.0) |
| Graduate or post-graduate Degree | **9.0** | **46.0** | 51 (27.5) |
| **OCCUPATION** | | | |
| Student | 32.0 | 35.0 | 67 (33.5) |
| Homemaker or domestic employee | 30.0 | 21.0 | 51 (25.5) |
| Employed or salesman | 28.0 | 38.0 | 66 (33.0) |
| Unemployed or Other | 10.0 | 6.0 | 16 (8.0) |
| **MARITAL STATUS** | | | |
| Single | 35.0 | 41.0 | 76 (38.0) |
| Married or living with someone | 60.0 | 54.0 | 114 (47.0) |
| Divorced | 1.0 | 2.0 | 3 (1.5) |
| Free union/ Widower | 16.0 | 10.0 | 26 (13.0) |
| **PREVIOUS DIAGNOSIS OF CHRONIC DISEASE** | | | |
| Diabetes | **10.0** | **2.0** | 12 (6.0) |
| Hypertension | 9.0 | 7.0 | 16 (8.0) |
| Overweight or Obesity | 26.0 | 25.0 | 51 (25.5) |
| **NUTRITION KNOWLEDGE** | | | |
| Knowing the mean daily energy requirements of an adult (1500–2000 kcal) | 11 | 7.0 | 18 (9.0) |
| **GDA ASPECTS** | | | |
| I am aware of the existence of the GDA | 85.0 | 74.0 | 159 (79.5) |
| I read the GDAs when I purchase food* | 14.0 | 23.0 | 37 (18.5) |
| I spend more than 30 seconds reading the GDA* | 37.0 | 43.0 | 80 (40.0) |
| The GDA are nothing/ little understandable* | 38.0 | 20.0 | 58 (29.0) |
| I never or almost never consider the GDA during food purchasing decisions* | 46.0 | 27.0 | 73 (36.5) |

SES, Socioeconomic Status; GDA, Guideline Daily Amounts.

**Bolds** indicate differences between low- and middle-SES participants.

*Proportions indicate the percentage of participants of those aware of the existence of the GDA.

also questioned the meaning of the numbers and whether they were targeted towards children or adults. Additionally, participants could not easily tell how many individual portions were contained in a package. They mentioned they rarely read Guideline Daily Amounts as they did not know how to interpret this label.

## Healthy choices

This label was evaluated during Phase I. Participants mentioned it was visible and clearly identifiable on the products, and that it allowed identifying between healthy and unhealthy products. However, they questioned the credibility of who determines how "healthy" a food

product is, or the criteria underlying this classification. Additionally, given that this FOPL classifies food products as "healthy" or "unhealthy", participants highlighted the need to revise the nutrition facts panel if information on specific ingredients or nutrients is required.

> M: *"What could you tell me about this label?"*
>
> P: *"It would guarantee that it (the product) is healthy . . . but I would still need to do my job by making numbers to know how many calories I need . . ." (Males with children ages 3–12, medium SES)*

Regarding subjective understanding, participants reported this label was easy to understand because both, the "tick" and the text "Healthy Choice" on the logotype are related to the healthiness of the product.

## 5-Color Nutrition Labelling

This label was evaluated during Phase I. Participants were initially confused as they could not interpret this label at a glance:

> M: *"This (5-Color Nutrition Label) is another label, ¿how do you like it? ¿what does it tell you?"*
>
> P1: *"At a glance it is confusing, you need an explanation before . . ." (Males and females ages 13–15, medium SES)*

Regarding the labels on the real products, participants mentioned that the label was not easily identifiable due to its size, although colors were considered as a strength of the label. As for the subjective understanding of this FOPL, participants questioned the meaning of the colors and the letters of the label, and interpretations were frequently unrelated to the purpose of this FOPL:

> M: *"What do you think the letters mean?"*
>
> P2: *"They represent the vitamins that the product has" (Females with children ages 3–12 y, medium SES)*

Participants considered that this label had too many different elements (i.e. the colors, the size of the circles, and the letters) which did not seem to communicate a clear message. Consequently, they mentioned it was difficult to decide if a certain product was "healthy".

## Ecuador's Multiple Traffic Light

This label was evaluated during Phase I and II. Participants showed high acceptability for this FOPL as it was readily related with a traffic light. When showing the amplified boards and the food products, they mentioned that the label could be easily identified on the package because of the colors of this label, that even at the distance they could tell if a product was healthier than other without reading the text, and that this label allowed making healthy food choices rapidly.

Regarding subjective understanding, participants mentioned that the colors of this label represented clear messages, such as "Green is healthy, amber more or less healthy, and red, little healthy". They reported understanding that the higher the number of green bars, the healthier the product. On the contrary, the higher the number of red bars, the less healthy the food

product. The amber bar was interpreted as the half-way measure. Additionally, the text and the length of the colored bars reinforced the correct interpretation of this label and the ability to discern between the different ingredients contained in the food product (i.e. fat, sugar and salt). These testimonials were consistent across the different testing materials.

> M: *"Could you tell me the first thing that comes to your mind (when looking at this FOPL) . . .?"*
>
> P: *"You immediately identify the healthy (products): the one with more green bars and an amber one. The one with two red bars and a green bar stands out . . ." (Males and females 21–23 y, low SES)*

## Health Star Rating

This label was evaluated during Phases I and II. Initially, participants were able to easily identify this label on food products, however, they mentioned it was difficult to interpret. After a while, participants apparently managed to understand the label through the interpretation of the number of stars. They reported liking the Health Star Rating because they could choose among a variety of products ranging in nutritional quality or "healthiness".

> M: *"What does it (the Health Star Rating) tell me? What do I understand?"*
>
> P: *"I like it because it does not force you to buy the healthiest, because sometimes these products are the most expensive. You can choose the second-best option (Males with children ages 3–12, low SES)*

As mentioned above, participants were able to understand the Health Star Rating by interpreting the number of stars. Other reported elements relevant for the understanding of this label were the ranking number and the colored filling bar. Additionally, they reported understanding that "the more stars, the healthier the product". However, the text included in this label (i.e. *Indice de Estrellas Saludables* or Health Star Rating in English) was not mentioned as an element improving the understanding of the label.

## Warning labels

This label was evaluated during Phase I and II. During phase I, we tested the Chilean Warning labels (in black) (Fig 1). Participants mentioned that this type of label is visible and easily identified on the front of the pack because of its size and color. They considered this label as adequate as each Warning labels evaluates a specific nutrient, but they considered black as an unattractive color, and mentioned the label was disruptive as it prevented them from fully appreciating the "attractiveness" of the package. When showing the shelf poster, they disliked the fact that this FOPL highlights only nutrients-to-limit and mentioned they were forced to choose the least "harmful" given that most products had at least one Warning labels.

> M: *"Here are different products, how do you see them?*
>
> P: *Boring, I don't like it, whatever, it doesn't have any color, the same figure, everything is "High" and bad (Males and females ages 13–15, low SES)*

Participants expressed understanding that the purpose of this label was to identify and warn about the various "unhealthy" nutrients of the product and clearly related it to the health

risks inferred by consuming products with the labels. They expressed understanding at a glance the warning shape indicating the "high content" of "unhealthy" nutrients. It took a moment for participants to understand that the more labels, the less healthy the product.

During Phase II we tested the Warning labels in red, based on results from Phase I. Perceptions about the ability to identify this label on the food products was similar to those of the Warning labels in black, however this version was not considered as disruptive with the design of the front of the pack. Participants also mentioned red was easier to identify compared to black, because it was more attractive.

Red labels had a better subjective understanding compared to black ones. Participants easily identified that the Warning labels in red allowed them to classify as "unhealthy" those products with more labels or warnings on the front of the pack. For example, when presenting the shelf poster, participants mentioned they would first see the number of labels and then read the text. The red color was related with traffic signs, turning on an alert or a warning signal: the more warnings a product has, the stronger the alert.

M: *"So tell me, ¿how do you like this (Warning red)?*

P1: *"Stop eating it, that is why they are red"* (Males and females ages 55–70 y, medium SES)

P2: *"I would go with the products that only have one label"* (Males with children ages 3–12 y, low SES)

## Discussion

Our study provides new evidence on the subjective understanding and acceptability of seven FOPL formats currently in use in different countries among low- and middle-income Mexican consumers. Few previous studies have assessed these FOPL together. Our study shows that although 80% of participants were aware of the Guideline Daily Amounts, the current FOPL in Mexico, of these only 1 in 3 considered it understandable or used the label. Regarding acceptability and understanding of the other FOPL evaluated, the 5-Color Nutrition Label emerged as the least adequate, whereas no single system clearly emerged as the best solution among the Health Star Rating, the red Warning labels, and the Multiple Traffic Light.

In Mexico, Guideline Daily Amounts were implemented as part of an industry-led strategy to address the rise in overweight, obesity and other non-communicable diseases, with an important role of the food industry on the approval of this policy and regardless the lack of evidence of their effectiveness to improve healthy food choices among Mexican consumers [34–35] . Previous research in Mexican and Chilean consumers indicates that Guideline Daily Amounts is not an adequate FOPL to promote informed food choices and, in consequence, the selection of healthy foods among Latin-American populations [16, 17, 36, 37]. Our results confirm these findings by showing that despite their high awareness among participants, Mexican Guideline Daily Amounts are rarely used, and this label has poor subjective understanding and acceptability.

In our study, Ecuador's Multiple Traffic Light had a high subjective understanding, similar to the one reported for other directive (i.e. Warning labels in red) and semi-directive (i.e. Health Star Rating) formats. Previous evidence has shown that this FOPL system is the most effective in influencing consumers' purchasing decisions among European consumers [38]. Studies among Mexican consumers have also shown that Multiple Traffic Light may be effective in helping consumers identify foods with high nutritional quality [16]. In contrast, in Ecuador, a qualitative study of consumer perceptions of the Multiple Traffic Light conducted

three years after its implementation, reported that although participants were aware of the label and reported understanding the message it conveyed, not all used the label for making purchasing decisions or had changed their attitudes and practices related to the purchase and consumption of processed foods [38, 39]. Based on this and other results showing little changes in consumption patterns among the Ecuadorian population after the implementation of the Multiple Traffic Light, it has been recommended to adapt and strengthen the label [38, 39]. It has been suggested that although as Multiple Traffic Light may increase the accuracy of perceptions of nutrient contents, they may not be more effective in discouraging the selection of products with excessive content of critical ingredients as they create decisional conflicts, especially among products with a combination of colors (e.g. green, amber and red) [40, 41]. Given the high acceptability and subjective understanding of the labels, future studies should explore the effectiveness of this label in real-life settings.

The potential of graded labels to guide consumers towards healthier food choices has also been emphasized in recent years [42, 43]. Interestingly, in our study participants had different reactions towards the two graded labels included (i.e. the Health Star Rating and the 5-Color Nutrition Label). In our study participants reported a low subjective understanding for the 5-Color Nutrition Label. This may be explained by the fact that in Mexico letters are rarely used as an evaluation method; rather, grading systems are generally based on a 1–10 scale, making it difficult for consumers to relate the letters of the label with the nutritional quality of the food product. Our results may be considered as contrasting with a previous study including Mexican consumers which showed that the Nutriscore, the final graphical format of the 5-Color Nutrition Label, had the best objective understanding [21]. However, differences may be due to the variations in the graphic design of the labels and the sampling frame, since 75% of Mexicans included in the previously mentioned study had an undergraduate or graduate degree. In contrast, 75% of participants in our study had a high school degree or lower, which resembles the most prevalent education level in the country [44]. These sub-groups are generally more at risk of consuming a lower-quality diet, as shown by a higher proportion of participants with a previous diabetes diagnosis among low-SES participants, and in consequence should be a target in any FOPL or other nutritional policies.

On the other hand, the Health Star Rating, a number-based graded label, had good acceptability and subjective understanding. Most elements of the label (i.e. the ranking number, the stars, and the colored filling bar) helped communicate that the more stars, the healthier the product. Of note is the fact that the text on the Health Star Rating was not helpful when interpreting the label, despite the translation was faithful to the meaning in English. Due to the high prevalence of non-communicable diseases in Latin-American countries and Mexico [5], graded formats evaluating the overall nutritional quality of a food product may not allow consumers identify high contents of critical nutrients. Results observed for the Healthy Choices logo, also evaluating the overall nutritional quality of a product, support this hypothesis. In line with a previous study in Mexican consumers showing that health logos were the easiest to understand [16], the Healthy Choices logo had a good subjective understanding. However, participants felt that this label did not provide enough information to make informed food choices. This may be explained by the fact that 41.5% of participants had been previously diagnosed with diabetes or hypertension, and consequently, were more interested in knowing the content of specific nutrients that have been identified as critical for chronic diseases (e.g. sodium, saturated fat). Studies comparing the effects of nutrient-specific and overall-rating labels should may help better understand the effects of these labels on the consumption of critical ingredients. Altogether, these findings highlight the importance of targeting the design of a FOPL towards specific populations, as epidemiological and cultural factors may determine the effectiveness of a specific label [45, 46].

In recent years, Warning labels have emerged as labels with a better potential in addressing chronic diseases given their ability to discourage consumption of products with high contents of critical ingredients [47]. Recent studies in Latin-American populations indicate that Warning labels improve the ability to correctly identify products containing excessive amounts of critical nutrients among adults[17] and are more effective in discouraging the selection of foods of low nutritional quality among children [18]. In line with these results, our study shows that the understanding of this label, as reported by participants, is very straightforward as participants were able to identify that the label aimed at warning about critical nutrients that could represent a health risk. However, the original version of this FOPL had a low acceptability among the labels tested. Research on the use of warning labels on tobacco products suggests that the most effective labels are those which include images that elicit a strong negative emotional response [48–50]. Consistent with this, in the present study, Warning labels were the only labels producing negative emotional responses. A study among British parents of children studying the effect of Warning labels on the selection of sugar-sweetened beverages reported that the effect of the labels was mediated by the elicitation of higher levels of negative emotional arousal, and therefore the most effective Warning labels were those making the threat more explicit [51]. The successful implementation of an intervention partially depends on its acceptability by the public and policy makers [52], and therefore identifying a FOPL that is both effective and acceptable is important. Our results indicate that although black Warning labels were less accepted, red Warning labels could have a better acceptability as they are interpreted as a warning instead of a harmful message. This is consistent with studies in Chile and Canada supporting the acceptability of these labels [53, 54]. Efforts to evaluate the effect of this label in dietary consumption among Chileans are being conducted. Additionally, other countries have implemented already Warning labels FOPLs (i.e. Perú, Uruguay) and others are developing their Warning labels systems (México, Canada, Ecuador and Brazil) [10]. Based on our results indicating decreased acceptability when showing the poster with a high proportion of products with Warning labels, special attention should be given to possible desensitization towards the label.

To our knowledge, this study provides novel information on the acceptability and understandability of recent FOPL systems among Latin-American consumers, using focus groups in population of low and middle SES. We tested labels on real products, including other package elements, such as graphic elements or nutrition and health claims, which turned out to be relevant for label acceptability. However, some limitations should be highlighted. First, we tested the subjective understanding of the labels, which has been shown to overestimate the objective understanding [19]. Second, the sample is not representative of all Mexicans, however characteristics resemble those of Mexican adults and therefore results may represent a wider population. Third, this study was not conducted in a real purchasing situation, where participants generally have limited time to purchase their foods, can choose among a variety of products and are exposed to different distractions. Fourth, given the nature of the focus groups technique, during the sessions some participants may be influenced by other more dominant peers. However, during the development of the group, participants were encouraged to contribute individually to the discussion to mitigate the influence. In addition, we selected a preliminary graphical format of the Nutri-score label, which does not contain any text compared to the rest of the labels. Although this may partially explain the reduced label acceptability and understanding, we believe that our results are comparable to other studies testing the 5-Color Nutrition Label [21]. We also used simple and non-mixed labeling formats, which is why we selected the Health Star Rating label. Results related to this label could differ when using the complete Health Star Rating + Guideline Daily Amounts format.

## Conclusions

FOPLs, along with other nutrition policies, could contribute substantially towards the reduction of obesity and chronic disease at the population level. Our results support previous evidence indicating that Guideline Daily Amounts is not adequate for improving healthy food, as they have a poor understanding [16, 36, 37]. Given the epidemiological profile of the Latin-American region, with non-communicable diseases and obesity as the leading causes of death and morbidity, urgent actions are required to address this public health problems. Other more directive or semi-directive labeling formats, such as the Warning labels or the Multiple Traffic Light, could be more effective in helping the most vulnerable consumers make healthier food choices, particularly decreasing consumption of critical ingredients. Future studies should explore the effect of these labels on the actual purchasing behavior of Latin-American consumers [55], as well as the need of parallel mass-communication campaigns to improve label understanding and incentivize consumers to improve their food choices.

## Supporting information

**S1 Fig. Materials utilized.**
(DOCX)

**S1 Table. Characteristics of the participating groups/sessions.** SES, socio economic status; M, male; F, female.
(DOCX)

**S2 Table. Nutritional content and classification of products utilized.** 5-CNL, 5 Color Nutritional Labelling; MTL, Multiple Traffic Light; GDA, Guide Dairy Amounts; HSR, Health Ranting Stars.
(DOCX)

**S1 File. Screening instrument for eligibility.**
(DOC)

**S2 File. Topics guide for front of pack labels focus group.**
(DOCX)

**S3 File. Self-administered questionnaire.**
(DOCX)

**S4 File. Coding guide for focus groups transcriptions.**
(DOCX)

**S5 File. Results of the double-entry matrix coding.**
(DOCX)

## Author Contributions

**Conceptualization:** Alejandra Jáuregui, Simón Barquera.

**Data curation:** Alejandra Contreras-Manzano.

**Formal analysis:** Jorge Vargas-Meza, Selene Pacheco-Miranda, Alejandra Contreras-Manzano.

**Funding acquisition:** Simón Barquera.

**Investigation:** Jorge Vargas-Meza, Alejandra Jáuregui, Selene Pacheco-Miranda, Alejandra Contreras-Manzano, Simón Barquera.

**Methodology:** Jorge Vargas-Meza, Alejandra Jáuregui, Selene Pacheco-Miranda, Alejandra Contreras-Manzano.

**Resources:** Alejandra Jáuregui.

**Supervision:** Jorge Vargas-Meza, Alejandra Jáuregui, Alejandra Contreras-Manzano.

**Validation:** Jorge Vargas-Meza, Alejandra Jáuregui, Alejandra Contreras-Manzano.

**Visualization:** Simón Barquera.

**Writing – original draft:** Jorge Vargas-Meza, Alejandra Jáuregui.

**Writing – review & editing:** Jorge Vargas-Meza, Alejandra Jáuregui, Selene Pacheco-Miranda, Alejandra Contreras-Manzano, Simón Barquera.

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
