## [Decision Letter · Decision Letter 0]

29 Aug 2019

PONE-D-19-21167

Front-of-pack nutritional labels: Understanding by low- and middle-income Mexican consumers

PLOS ONE

Dear Miss Jauregui,

Thank you for submitting your manuscript to PLOS ONE. After careful consideration, we feel that it has merit but does not fully meet PLOS ONE’s publication criteria as it currently stands. Therefore, we invite you to submit a revised version of the manuscript that addresses the points raised during the review process.

We would appreciate receiving your revised manuscript by Oct 13 2019 11:59PM. To enhance the reproducibility of your results, we recommend that if applicable you deposit your laboratory protocols in protocols.io, where a protocol can be assigned its own identifier (DOI) such that it can be cited independently in the future. For instructions see: http://journals.plos.org/plosone/s/submission-guidelines#loc-laboratory-protocols

We look forward to receiving your revised manuscript.

Kind regards,

Robert Siegel

Academic Editor

PLOS ONE

Journal Requirements:

3. Please provide additional details regarding participant consent. In the ethics statement in the Methods and online submission information, please ensure that you have specified (1) whether consent was informed and (2) what type you obtained (for instance, written or verbal). If your study included minors, state whether you obtained consent from parents or guardians.

4. When reporting the results of qualitative research, we suggest consulting the COREQ guidelines: http://intqhc.oxfordjournals.org/content/19/6/349. In this case, please consider including more information on the moderator (training and characteristics), and whether risk of bias was assessed.

5. We note that Figure S1 in your submission contain copyrighted images. All PLOS content is published under the Creative Commons Attribution License (CC BY 4.0), which means that the manuscript, images, and Supporting Information files will be freely available online, and any third party is permitted to access, download, copy, distribute, and use these materials in any way, even commercially, with proper attribution. For more information, see our copyright guidelines: http://journals.plos.org/plosone/s/licenses-and-copyright.

1.    You may seek permission from the original copyright holder of Figure S1 to publish the content specifically under the CC BY 4.0 license.

6. We note that you have indicated that data from this study are available upon request. PLOS only allows data to be available upon request if there are legal or ethical restrictions on sharing data publicly. For information on unacceptable data access restrictions, please see http://journals.plos.org/plosone/s/data-availability#loc-unacceptable-data-access-restrictions.

Additional Editor Comments:

A very interesting manuscript that addresses several important issues. 

Reviewers' comments:

Reviewer's Responses to Questions

**Comments to the Author**

1. Is the manuscript technically sound, and do the data support the conclusions?

Reviewer #1: Yes

Reviewer #2: Partly

2. Has the statistical analysis been performed appropriately and rigorously? 

Reviewer #1: Yes

Reviewer #2: I Don't Know

3. Have the authors made all data underlying the findings in their manuscript fully available?

Reviewer #1: Yes

Reviewer #2: No

4. Is the manuscript presented in an intelligible fashion and written in standard English?

Reviewer #1: Yes

Reviewer #2: No

5. Review Comments to the Author

Reviewer #1: Review of “Front-of-pack nutritional labels: Understanding by low- and middle-income Mexican Consumers” for PLOS One

Summary

Our main objective was to explore the subjective understanding of labels that are currently used internationally among low- and middle -income Mexican consumers.

We performed two phases of 10 focus groups with adolescents (13-15 y), young adults (21-23 y), mothers of children 3-12 y, fathers of children 3-12 y and older adults (55-70 y). Seven FOPL were evaluated: Guideline Daily Amounts (GDA), Multiple Traffic Light (MTL), Chilean Warning labels (WL), WL in Red, 5-Color Nutrition Label (5-CNL), Health Star Rating (HSR), and Healthy Choice label. Data was analyzed with a triangulation of researchers using a content analysis, based on three codes: 1) awareness and use of the GDA, 2) acceptability, and 3) subjective understanding of labels

Major comments

Qualitative analyses with sufficiently large group (200 interviews) and appropriate for this type of analysis:

“Audios from focus groups were transcribed verbatim. The systematization and interpretation of the information was guided using the content analysis technique using Atlas.Ti version 11 [29,30]. We developed a priori central themes for our research questions before data was collected. Additionally, we also 218 developed a code catalog based on transcriptions. The analysis was based on three codes: 1) awareness and use of the GDA, 2) acceptability, and 3) subjective understanding of labels. Double entry conceptual matrices were developed for each code and participant group. Information was encoded and analyzed independently by the focus group moderator, as well as by two additional researchers (JVM and SPM), one of whom attended focus groups as listener.”

Timeliness of paper given the statement (lines 81-82): Thus, the regulation of FOPL in the country is currently being revised, to replace this system with one that better serves the purpose of decreasing sugar, salt and sodium intakes among the Mexican population.

Contribution: focus on lower income and education groups, which are the most representative of the Latin-American populations and generally the most nutritionally at-risk

200 participants (100 in each phase)

Minor editing

Data is plural-- change throughout ie Data was were analyzed with a triangulation of researchers using a content analysis, based on three codes

Line 65: insert which are before considered critical nutrients for chronic diseases

Line 92: nutritional quality of products, such as HSR and Nutriscore, may allow consumers to identify

Line 103 In order to contribute to fill this gap in literature

Line 192: facilitator did not explain to participants the intended message of the labels

Line 235: foods (Table 1). Among those reading the GDA, 37% considered the label as incomprehensible or minimally understandable

Line 309: They considered that it had too many confusing elements that made it difficult to interpret, and consequently they were not able to identify if the products were healthy.

Health Star Rating—this section is confusing –rewrite

Line 395: Mexican consumers. These FOPL have been little assessed all together in previous studies. Rewrite as Few previous studies have assessed these FOPL together

Reviewer #2: Overall, I like this manuscript. The research is on an important topic: getting low and middle SES Latin American individuals to make healthier food selections than they otherwise would via information on the food packaging. The general research methodology is clearly described (the photos really helped) and appropriate to achieve the study aims. The paper is interesting to read and well-written. I have some suggestions to improve the rigor of the analysis and the clarity of the paper.

1) My biggest concern with the manuscript is that the authors have not provided analysis results and/ summary tables from their qualitative analysis of the focus group interviews.

a) For example, I was expecting there to be a table with the major themes that emerged, and the number of participants or comments that supported that theme. Instead, the authors just state their conclusions from the focus groups without providing the reader any evidence that their conclusions are valid.

b) The authors could provide the list of questions asked during the focus groups (perhaps as an appendix), and their data collection instrument that was mentioned on line 119.

c) On line 224, you state that the researchers reached similar conclusions. Do you have statistical analysis or some kind of way of quantifying this claim, such as kappa scores indicating agreement?

d) On line 225, you state that "results on analytics codes are presented for each of the FOPL tested." Where are these results presented? Do you mean the descriptions that follow later in the main body of the paper? I would expect these results to include quantitive statements, something like: 18 out of the 20 people who mentioned the Mexican GDA stated that they did not find it helpful. Instead the authors write (on lines 251-253, "Food purchasing decisions were generally based on price... whereas GDA was generally not mentioned as a purchasing criteria." And later on in that paragraph, "Older adults also tended to report using GDA more frequently for health reasons." It is those types of statements that could be more rigorous by providing specific counts and agreement percentages. I imagine this information be contained in a table. This comment applies to all of the food labels that are discussed (Healthy Choices, 5-CNL, MTL, HSR, WL.

2) I was not familiar with some of the terms used in the manuscript. The authors could consider using different terms or defining them.

a) Free sugars (as opposed to just "sugars")

b) Acceptability. What does this mean with regard to food labels? Please define or use a different word that is more obvious in its meaning.

What is a "Gessell chamber"? Why does it matter for your study? If it is important, please describe what it is and why it helps your research to have had the focus group sessions take place in Gessell chambers.

3) The paper would benefit from a careful copy-editing.

a) There are a few places that seem to be missing commas (introductory phrases such as "In Mexico,.." on page 3 introduction, line 60 and again on page 4, line 78. In June 2016, On page 4, line 71.

b) There are a few places where the grammar is slightly off.

i) Line 460 "participants considered this label did not provide..." I think the sentence would read better if you either inserted the word "that" after considered, or better yet, rewrote it to be: "participants felt that this label did not provide.."

ii) Line 462 "in consequence" should be "consequently"

iii) Line 504 "which has been proved to overestimate" should be "which has been shown to.."

iv) Line 506 "characteristics resemble to those of Mexican..." Delete "to" in that phrase.

v) Line 508 "have a limited time to purchase". I suggest deleting "a".

4) In future research, you mention testing the labels with actually food purchases. There is prior research that does investigate actual food purchases as impacted by labels, although perhaps not in Latin America. For example, Mazza, M. C., Dynan, L., Siegel, R. M., & Tucker, A. L. (2018). Nudging Healthier Choices in a Hospital Cafeteria: Results From a Field Study. Health Promotion Practice, 19(6), 925–934. https://doi.org/10.1177/1524839917740119. investigates the use of traffic light on food purchases in a workplace cafeteria.

I wish the authors the best of luck in revising their manuscript.

6. PLOS authors have the option to publish the peer review history of their article (what does this mean?). If published, this will include your full peer review and any attached files.

Reviewer #1: No

Reviewer #2: Yes: Anita L. Tucker

---

## [Author Response · Author response to Decision Letter 0]

22 Oct 2019

All responses have been uploaded in the attached Response to Review (Rebuttal letter).

---

## [Decision Letter · Decision Letter 1]

1 Nov 2019

Front-of-pack nutritional labels: Understanding by low- and middle-income Mexican consumers

PONE-D-19-21167R1

Dear Dr. Jáuregui,

We are pleased to inform you that your manuscript has been judged scientifically suitable for publication and will be formally accepted for publication once it complies with all outstanding technical requirements.

With kind regards,

Robert Siegel

Academic Editor

PLOS ONE

Additional Editor Comments (optional):

The reviewers and I agree that the manuscript should be accepted. Reviewer two asked some minor edits which are very appropriate.

Reviewers' comments:

Reviewer's Responses to Questions

**Comments to the Author**

1. If the authors have adequately addressed your comments raised in a previous round of review and you feel that this manuscript is now acceptable for publication, you may indicate that here to bypass the “Comments to the Author” section, enter your conflict of interest statement in the “Confidential to Editor” section, and submit your "Accept" recommendation.

Reviewer #1: All comments have been addressed

Reviewer #2: All comments have been addressed

2. Is the manuscript technically sound, and do the data support the conclusions?

Reviewer #1: Yes

Reviewer #2: Yes

3. Has the statistical analysis been performed appropriately and rigorously? 

Reviewer #1: Yes

Reviewer #2: I Don't Know

4. Have the authors made all data underlying the findings in their manuscript fully available?

Reviewer #1: Yes

Reviewer #2: No

5. Is the manuscript presented in an intelligible fashion and written in standard English?

Reviewer #1: (No Response)

Reviewer #2: Yes

6. Review Comments to the Author

Reviewer #1: (No Response)

Reviewer #2: I have a few minor comments.

1) I continue to find it difficult to remember all of the acronyms that you use. I think it would increase readability if you spelled out the front-of-package labels that you investigate: MTL, WL, HSR, 5-CNL, and GDA.

2) I believe that "intakes" should actually be singular. (See the abstract and other parts of the paper). Consider substituting the word "consumption" instead.

3) Line 215 -- add the word "to" between explaining and participants. (".. without explaining TO participants..")

7. PLOS authors have the option to publish the peer review history of their article (what does this mean?). If published, this will include your full peer review and any attached files.

Reviewer #1: No

Reviewer #2: No

---

## [Editor Report · Acceptance letter]

7 Nov 2019

PONE-D-19-21167R1 

Front-of-pack nutritional labels: Understanding by low- and middle-income Mexican consumers 

Dear Dr. Jáuregui:

I am pleased to inform you that your manuscript has been deemed suitable for publication in PLOS ONE. Congratulations! Your manuscript is now with our production department. 

With kind regards,

on behalf of

Dr. Robert Siegel 

Academic Editor

PLOS ONE